# The Impact of Spousal Migration on the Mental Health of Nepali Women: A Cross-Sectional Study

**DOI:** 10.3390/ijerph17041292

**Published:** 2020-02-17

**Authors:** Nirmal Aryal, Pramod R. Regmi, Edwin van Teijlingen, Steven Trenoweth, Pratik Adhikary, Padam Simkhada

**Affiliations:** 1Faculty of Health and Social Sciences, Bournemouth University, Bournemouth BH1 3LH, UK; pregmi@bournemouth.ac.uk (P.R.R.); evteijlingen@bournemouth.ac.uk (E.v.T.); strenoweth@bournemouth.ac.uk (S.T.); 2Chitwan Medical College, Tribhuvan University, Bharatpur P.O. Box No. 42, Nepal; 3Datta Meghe Institute of Medical Sciences, Wardha 442001, India; 4Manmohan Memorial Institute of Health Sciences, Tribhuvan University, Kathmandu P.O. Box No. 15201, Nepal; p.p.simkhada@hud.ac.uk; 5Nobel College, Pokhara University, Kathmandu GPO 10420, Nepal; 6School of Public Health, University of California, Berkeley, CA 94720-7360, USA; adpratik30@gmail.com; 7Green Tara Nepal, Kathmandu GPO 9874, Nepal; 8School of Human and Health Sciences, University of Huddersfield, Huddersfield HD1 3DH, UK

**Keywords:** mental health, migrant, left-behind, spouse, depression, resilience, Nepal, low- and middle-income countries

## Abstract

Spousal separation, lack of companionship, and increased household responsibilities may trigger mental health problems in left-behind female spouses of migrant workers. This study aimed to examine mental ill-health risk in the left-behind female spouses of international migrant workers in Nepal. A cross-sectional survey was carried out in the Nawalparasi district. Study areas were purposively chosen; however, participants were randomly selected. Nepali versions of the 12-item General Health Questionnaire (GHQ), Beck Depression Inventory (BDI), and Connor–Davidson Resilience Scale (CD-RISC) were used. Mental ill-health risk was prevalent in 3.1% of the participants as determined by GHQ. BDI identified mild or moderate depression in 6.5% of the participants with no one having severe depression. In bivariate analysis, a high frequency of communication with the husband was associated with lower mental ill-health risk and depression, as well as increasing resilience. Reduced return intervals of husbands and a high frequency of remittance were also associated with a low GHQ score. In a multiple regression model, adjusting for potential confounding variables, participants who communicated with their husbands at least once a day had a greater mean CD-RISC score (i.e., high resilience against mental ill-health risk) compared to those who did so at least once a week; a mean difference of 3.6 (95% CI 0.4 to 6.9), *P* = 0.03. To conclude, a low mental ill-health risk was found in the female spouses of migrants.

## 1. Introduction

Migration is a global phenomenon and particularly common in low- and middle-income countries (LMICs). Around 3.5% of global population, i.e., 272 million, were international migrants in 2019, of whom more than half (164 million) were migrant workers [1]. According to the World Bank report, international migrants sent a record high remittance of US$ 466 billion in 2017, which is three times higher than the official development assistance [2]. 

International labour migrant workers from Nepal have grown substantially in the recent years; mainly India, Malaysia, and the six countries of Gulf Co-operation Council (GCC) are the key destinations. The Ministry of Labour has issued labour permits for 3.5 million Nepali between 2008/09 to 2016/17, of which 86.4% were for Malaysia and the Middle East [3]. It is estimated that an additional one million Nepali work in India as temporary migrants who do not require a labour permit due to the free movement treaty between India and Nepal [4]. The economic contribution of Nepali migrants to the country is highly significant. In the fiscal year 2016/17, they sent over US$6.1 billion in remittances, which was 26.3% of Nepal’s total Gross Domestic Product (GDP) [5]. 

Although there are limited studies focusing on health and well-being of left-behind female spouses, they usually report a decreased health and well-being compared to non-migrant wives. For example, a study among left-behind female spouses in Mexico reported a higher prevalence of heart disease, overweight, and obesity [6]. A study in China found a lower score in health-related quality-of-life among left-behind female spouses than non-left-behinds [7]. An Indian study concluded that the migration of husbands does not improve the health of the left-behind wives, but instead puts them at a greater risk of reproductive morbidities [8]. In relation to the mental ill-health risk among left-behind female spouses, evidence is stronger towards an increased risk [9,10]. Female spouses of migrant workers may be vulnerable to mental ill-health risk because of the pain of spousal separation and lack of companionship; due to the increased household responsibilities, i.e., undertaking the work, which is mostly done by the husband in a male dominant society; and an increase in daily stressors or being worried about the health and well-being of the migrating husband. Globally, evidence on mental health well-being of female spouses of migrant workers is extremely scarce [11]. Despite the large number of female spouses of migrant workers who are left-behind, none of the studies have examined their mental ill-health risk in Nepal using validated mental health assessment tools. Against this backdrop, we hypothesized that the mental health of left-behind female spouses of migrant workers may be negatively affected, and that they have a higher mental ill-health risk than the general population.

## 2. Materials and Methods 

This was a cross-sectional study.

### 2.1. Study Setting and Sample

During April–May 2018, a survey was carried out among female spouses of Nepali migrant workers. Participants were from two rural municipalities, namely, Pratappur and Susta of the Nawalparasi district. The Nawalparasi district was purposively selected because it has one of the highest out-migration rates in the country with 106,693 labour permits issued between 2008 and 2009 to 2016 and 2017 [3]. Study participants were defined as “female spouses of Nepali migrants who were ‘left-behind’ for at least six months in the recent past”.

### 2.2. Sampling Process

The study areas (the Pratappur and Susta rural municipalities) were purposively selected based on the possibility of recruiting enough samples and the feasibility to conduct the study. In the second stage, five wards (local small administrative units) from each of these rural municipalities were randomly selected. A sampling frame of the eligible participants in those areas was developed and a unique number was assigned to them. Finally, participants were randomly selected by a computer-based randomization technique. If the participants were not at home or did not want to participate in the study, an eligible participant nearby was selected. The overall response rate was 94%. Non-response was mainly due to (a) time limitation and (b) unwillingness to take part.

The total required sample size was estimated to be 359 after adjusting for design effect 2 and a 10% non-response rate, which will give a 5% margin of error for the proportion of depression in the Nepali population of 11.7% [12].

### 2.3. Data Collection Procedures

Standardized questionnaires were adapted to collect different aspects of the socio-demographic characteristics, work and living conditions, and health- and behaviour-related information; for example, the Nepal Demographic Health Survey and the World Health Organization (WHO) STEPS survey. A Nepali version of the 12-item General Health Questionnaire (GHQ) was used to identify minor psychiatric morbidities [13]. Severity of depression was measured by the Nepali version of the Beck Depression Inventory (BDI) [14]. Resilience of the participants was assessed using the Nepali version of the Connor–Davidson Resilience Scale (CD-RISC) [15]. All these tools have been previously validated in Nepali. 

The survey was carried out by experienced local female researchers. Two research assistants received week-long training around survey data collection, which was facilitated by an author P.R.R. 

### 2.4. Statistical Methods

Descriptive statistics were generated using the mean, standard deviation (SD), frequency, and percentages. The data from the GHQ-12 questionnaire was analysed using a dichotomous score for each of the 12 questions (i.e., scores of 0 and 1 were considered “0” and that of 2 and 3 were considered “1”). Participants with an overall score of equal to or greater than 6 were defined as having a high mental ill-health risk [16] and thus was dichotomized to identify those with minor psychiatric morbidities. The BDI score was categorized as having no depression (0 to 13), mild depression (14 to 19), moderate depression (20 to 28), and severe depression (29 to 63) [17]. The CD-RISC was presented as a mean score. Age and completed education years were included as confounding variables whereas important migration-related variables were also included in the model to estimate the association of the mean CD-RISC score (response variable) and “communication with husband” variable (explanatory variable) in a multiple linear regression model. We performed data analysis using STATA software version 14 (Stata Corporation, College Station, TX, USA).

### 2.5. Patient Public Involvement

During the conception phase, we have consulted with left-behind female spouses (=lay experts) and people from migrant-related organizations in Nepal regarding the rationale of this study, the scope, and possible questions. A pilot test [18] was carried out in the study areas prior to the data collection to assess if employed tools/contents are comprehensive and culturally accepted by the prospective participants. 

### 2.6. Ethics Approval and Consent to Participate

The study protocol was approved by the Research Ethics Committee (REC) of Bournemouth University, UK (Ref 20234) and the Ethical Review Board (ERB) of the Nepal Health Research Council, Nepal (Ref 2286). The present study was conducted in compliance with all human rights and ethical standards required by health researchers conducting studies among human subjects on sensitive issues [19]. All survey procedures were designed to protect participants’ privacy, allowing for anonymous and voluntary participation. Written informed consent was obtained from all respondents. Through an information sheet in the Nepali language, respondents were provided with enough information about the study procedure, confidentiality, study purpose, risk and benefits, complaint procedure, etc. No monetary incentives were provided to the participants.

## 3. Results

A total of 382 female spouses of migrant workers participated in this study. Table 1 shows the key socio-economic characteristics of the participants. Very few participants had their own occupation and of them most were running shops or were tailors. In contrast with the national figure in ethnicity, which has dominance of *Brahmin* and *Chhetri* [20], more than two-thirds of the participants were from indigenous *Terai* and *Madhesi* ethnicity. The vast majority (91.9%) of the participants had children (mean 1.4, SD 0.7). Around 63.9% of the participants were living in a joint family. Use of social media was very common among participants; 79.3% used social media at least once a week and more than two-thirds (67.8%) had their own social media account.

Migration-related characteristics of participants’ husbands are shown in Table 2. Malaysia and Saudi Arabia were the major destination for husbands of participants and most of them were involved in technical and manual labour. The majority of the participants communicated with their husband at least once a day and almost all (96.8%) reported doing so using internet-based applications (e.g., Viber, Imo, and Facebook).

Consumption of tobacco and alcohol was very rare among the participants. Self-reported prevalence of ever user of tobacco and alcohol was 0.5% and 0.3%, respectively. Almost all participants (98.9%) performed moderate physical activity every day whereas 59.8% reported being involved in vigorous physical activity on 4.6 days (SD 1.6) of the week on average. Overall, 9.4% of the participants were currently on medication for any medical condition. Nearly two-thirds (61.5%) of the participants reported that they usually decide themselves about their health care. 

Only around 3.1% ever had a suicidal ideation and 1.3% self-reported having attempted it. Mental ill-health risk was prevalent in 3.1% of the participants as determined by GHQ. BDI identified mild or moderate depression in 6.5% of the participants with no one having severe depression. The mean GHQ, BDI, and CD-RISC scores were 8.3 (SD 4.1), 6.3 (SD 4.6), and 76.5 (SD 13.9), respectively. Participants aged equal to or above 30 years were more likely to have higher mean scores of GHQ and BDI compared with those aged below 30 years (Figure 1), and these differences were statistically significant; a mean difference of 1.1 (95% CI 0.2 to 1.9), *P* = 0.01, for GHQ, and 1.0 (95% CI 0.1 to 1.9), *P* = 0.03, for that of BDI was calculated via Student’s t-tests. This trend was similar for the CD-RISC score, but the difference was not statistically significant. There were no significant correlations of number of children with mean GHQ, BDI, and CD-RISC scores. However, number of children was significantly and negatively correlated with age, r = −0.2, *P* < 0.001.

The details of the bivariate associations of the mean GHQ, BDI, and CD-RISC scores with key migration-related variables are given in Table 3. The high scores of the GHQ and BDI represent an increasing risk whereas that for the CD-RISC indicates increasing resilience against mental ill-health risk. A high frequency of communication with husbands was associated with low mental ill-health risk (low GHQ mean score), depression (low BDI mean score), and increasing resilience (high CD-RISC mean score). Reduced return intervals of husbands and a high frequency of remittance were also associated with a low GHQ score. 

Table 4 shows the association of the mean CD-RISC score (response variable) and “communication with husband” (explanatory variable) in a multiple linear regression model, adjusting for other potential confounders. Participants who communicated with their husbands at least once a day had a greater mean CD-RISC score compared to those who did so at least once a week; mean difference of 3.6 (95% CI 0.4 to 6.9), *P* = 0.03. Sensitivity analysis excluding outliers yielded consistent results.

Participants had a good knowledge and belief on mental illness and factors associated with it (Appendix A: Table A1). Most of them agreed that mental illness is quite common, triggered by stressful events and abuse, and can be treated. The vast majority disagreed that mental illness is a “shame” or caused by the “sins” of the past life.

## 4. Discussion

The prevalence of mental ill-health risk was 3.1% and mild or moderate depression was 6.5% in this study population of migrants’ female spouses. In bivariate analysis, communication frequency with the migrant husband, in-bound remittance frequency, and husband return frequency were mainly associated with the measures of mental ill-health risk. Migrants’ female spouses aged 30 years or above had significantly greater GHQ and BDI scores and lower CD-RISC score compared to younger counterparts. After adjusting for important confounding variables in a multiple regression model, communication with the husband at least once a day was strongly associated with increased resilience against mental ill-health risk. 

Our result on prevalence of mental ill-health risk and depression in migrants’ female spouses is in sharp contrast with the preponderance of existing evidence. Other studies usually reported higher mental ill-health risk in female spouses of migrants than ours [9,10]. For example, a Sri Lankan study among 277 spouses of international migrant workers reported prevalence of common mental disorders in 14.4% and depression in 12.3% [9]. A number of studies also suggested higher mental ill-health risk in female spouses of migrants compared to that of non-migrants [6,7,8,10,21,22]. The finding of the present study is also lower compared to the general Nepali population although using different tools. A national representative study carried out in Nepal reported the prevalence of anxiety and depression in 22.7% and 11.7%, respectively, among population aged 18-65 years [12]. Another sociological research carried out in Eastern Nepal concluded that increased workload triggered mental tension in female spouses of migrants [23].

Very few studies reported lower or non-significant differences in mental ill-health risk between female spouses/family members of migrants and non-migrants. A recent global study using Gallup World Poll data, including 144,003 participants from 114 countries (including Nepal), found that although higher depression and stress levels were pronounced in relatives of migrants from developed countries, it was just opposite in middle-income and LMICs with higher emigration rates [24]. A large longitudinal study in Mexico [25] and a small study in Iran [26] also did not find significant differences in mental ill-health risk between female spouses of migrants and non-migrants.

A comparatively low mental ill-health risk in the present study may have several possible explanations. First, more than two-thirds of the participants communicated with their migrating husband on a daily basis. This might have contributed to offset the stress of spousal separation and provide an opportunity to exchange the feelings and problems. Second, participants showed good knowledge and belief on mental illness and factors associated with it, which could have helped to mitigate stress and enhance resilience. Third, nearly half of the participants’ husbands were involved in semi-skilled work (e.g., technician, driver, and chef) who not only earn good money compared to labour workers but also have relatively low occupational risk and vulnerabilities. Thus, migrant spouses might have benefitted from both good remittance and worry less about the health and well-being of their husbands. Finally, out-migration is a historically common social phenomenon of Nepali societies to generate household income. Seasonal cross-border migration to India for work dates back two centuries [4] and there has been migration out-flux to the countries of GCC and Malaysia since the 2000s [3]. It is plausible that participants could have witnessed or experienced the impact of male migration even before marriage and possibly could be prepared to tackle consequent social and psychological effects from their husband’s migration. Family support for the left-behind female spouses in Nepali society depends on a number of factors, such as, either they live in a single or joint family, the personal relationship of the migrating husbands and their spouses with family members, the educational level of the family, and partly also on how well the migrating husband is earning overseas and sending back remittance.

A novel finding from this study is the importance of communication with the migrating husband for mental health well-being, which has not been documented before. Research on the impact of family communication on the mental health of the spouse is scarce. Available limited studies indicate a negative relationship between family communication and mental ill-health risk in family members, including spouses [27,28]. The increasing access to internet in LMICs and a rise in the use of social media create a good opportunity to explore this issue further. 

As far as we are aware this is the first study to identify mental ill-health risk among left-behind female spouses of migrants in Nepal and the first to examine resilience against mental ill-health risk in spouses of migrants worldwide. A limitation of this study includes participants were only from rural municipalities of low-land *Terai*. We acknowledge that results could be different in female spouses of migrants in urban areas of *Terai* as well as that of hilly and mountain regions due to the discrepancies in socio-economic status, knowledge of mental health, and skills of migrating husbands. There also could be the possibility of “social desirability bias” because mental ill-health is still a “taboo” in rural Nepali society and often synonymized as “insanity”, which might refrain participants to admit mental ill-health symptoms. We could not compare mental ill-health risk between female spouses of migrants and non-migrants, which could provide broader insights into this relationship. Because the study was carried out in one district of Nepal, findings should not be extrapolated to the country as a whole. Cross-border migration to India is very common in the study district. We did not include participants whose husbands were working in India because they were mostly seasonal migrants and return back home more frequently. Although we used validated tools, one should be aware of the validity of self-reporting owing to the low educational level of the participants.

## 5. Conclusions

The findings of the present study indicate an overall low mental ill-health risk in the female spouses of migrants. Communication frequency with the husband was associated with lower general mental ill-health risk and depression, and greater resilience. More evidence from larger and longitudinal studies from migrant-sending countries is required to understand the factors affecting the differences in mental ill-health risk in left-behind female spouses. 

## Figures and Tables

**Figure 1 ijerph-17-01292-f001:**
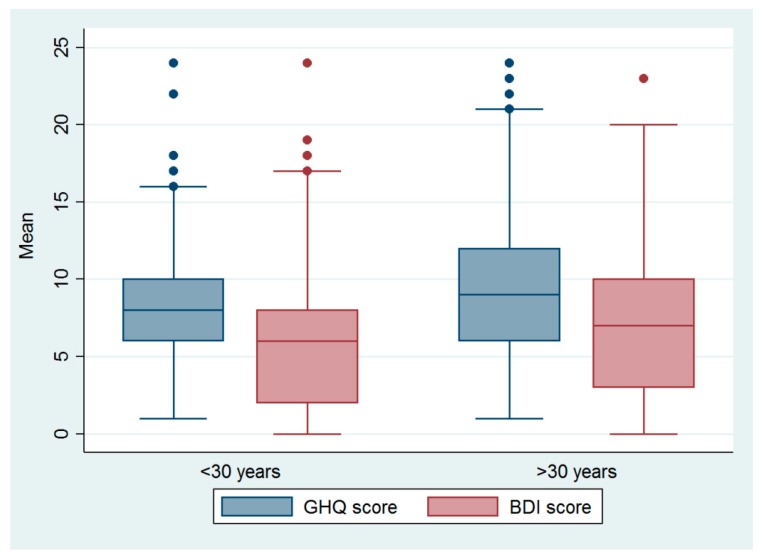
Mean General Health Questionnaire (GHQ) and Beck Depression Inventory (BDI) scores of participants aged below and above 30 years.

**Table 1 ijerph-17-01292-t001:** Socio-economic characteristics of the participants.

Variables	Mean (SD)
Age	29.8 (6.1)
Completed education (years)	8.1 (4.1)
Household income per year (NPR)	23,002 (7189)
Household members	5.9 (3.2)
	**Number (%)**
Ethnicity	
Indigenous Terai	168 (44)
Madhesi	102 (26.7)
Dalit Terai	39 (10.2)
Indigenous Hill	32 (8.4)
Brahmin/Chhetri/Thakuri	18 (4.7)
Muslim	18 (4.7)
Other	5 (1.3)
Religion	
Hindu	348 (91.1)
Muslim	21 (5.5)
Boudhha	12 (3.1)
Other	1 (0.3)
Social media account	300 (67.8)
Bank account	259 (67.8)
Property ownership	65 (17.0)
Occupation	27 (7.1)
NPR: Nepali Rupee, 1 US$=114 NPR (as of 10 February 2020)	

**Table 2 ijerph-17-01292-t002:** Migration related characteristics of participants’ husbands.

Variables	Number (%)
Destination country	
Malaysia	133 (34.8)
Saudi Arabia	108 (28.3)
Qatar	71 (18.6)
United Arab Emirates	51 (13.3)
Other	19 (5.0)
Occupation	
Technician	133 (34.8)
Labourer	109 (28.5)
Driver	40 (10.5)
Security guard	23 (6.0)
Hotel worker	21 (5.5)
Cook/chef	11 (2.9)
Agriculture worker	9 (2.4)
Shop/supermarket worker	8 (2.1)
Other	28 (7.3)
Average interval of return	
Within 1 year	16 (4.2)
1 to 2 years	157 (41.2)
2 to 3 years	159 (41.7)
3 to 4 years	38 (10.0)
Five years or more	11 (2.9)
Average frequency of remittance	
Once a month	96 (25.2)
Every three months	266 (69.8)
Once in every six months	16 (4.2)
Once a year	2 (0.5)
Not sent yet	1 (0.3)
Communication with husband	
At least once a day	276 (72.2)
At least once a week	103 (27.0)
Once a month	3 (0.8)
Usually who initiate for communication?	
Husband	214 (56.0)
Equally both	133 (34.8)
Wife	35 (9.2)

**Table 3 ijerph-17-01292-t003:** Bivariate associations of mean GHQ, BDI, and Connor–Davidson Resilience Scale (CD-RISC) scores with migration-related variables.

Variables	GHQ	BDI	CD-RISC
	Mean (SD)		
Living status			
Single family	8.7 (4.3)	6.4 (4.9)	76.1 (14.1)
Joint family	8.1 (3.9)	6.2 (4.4)	76.7 (13.8)
Destination country			
Middle East (*n* = 133)	8.1 (4.4)	6.1 (4.8)	77.0 (13.3)
Malaysia (*n* = 244)	8.7 (4.4)	6.7 (4.3)	75.9 (14.8)
Occupation of husband			
Labourer (*n* = 82)	8.7 (4.4)	6.4 (4.9)	73.5 (14.9)
Technician (*n* = 127)	8.0 (3.7)	6.5 (4.9)	77.1 (13.7)
Driver (*n* = 40)	8.0 (4.3)	6.9 (4.2)	76.0 (13.9)
Husband return frequency			
Within 2 years	7.2 (3.5) ***	6.4 (4.1)	77.2 (15.2)
More than 2 years	9.3 (4.3)	6.2 (4.9)	75.9 (12.7)
Remittance frequency			
Once a month	7.8 (4.0) *	5.8 (5.1)	78.5 (11.3) ***
Every three months	8.4 (4.0)	6.4 (4.3)	76.5 (14.5)
Six months or more	10.2 (4.8)	7.7 (5.7)	67.9 (13.0)
Communication with husband			
At least once a day	8.1 (4.3) *	5.9 (4.6) *	77.7 (13.5) **
At least once a week	8.9 (3.5)	6.9 (4.6)	73.8 (14.4)

* *P* = 0.06 to 0.10, ** *P* = 0.01 to 0.05, *** *P* < 0.01; *N* = 382 unless stated otherwise; Student’s t tests, Mann–Whitney *U* tests, analysis of variance (ANOVA) tests, and Kruskal–Wallis tests were performed.

**Table 4 ijerph-17-01292-t004:** Association of the mean CD-RISC score with the “communication with husband” variable in a multiple linear regression model, adjusting for confounding variables.

Variable	Estimate (95% CI)	*P* value
Communication with husband (at least once a day compared to at least once a week)	3.6 (0.4 to 6.9)	0.03
Age (per decade older)	−0.2 (−2.9 to 2.4)	0.84
Completed education (years)	0.02 (−0.3 to 0.3)	0.86
Husband return frequency (within two years compared to more than two years)	−0.6 (−3.5 to 2.3)	0.69
Husband migration duration (years)	0.02 (−0.04 to 0.08)	0.53
Living status (joint family compared to single family)	0.5 (−2.7 to 3.7)	0.75
Remittance frequency (at least monthly compared to more than monthly)	−2.2 (−5.6 to 1.1)	0.19

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
