# Peer review of "The Impact of Spousal Migration on the Mental Health of Nepali Women: A Cross-Sectional Study"

_ijerph, 2020, doi:10.3390/ijerph17041292_

Round 1

Reviewer 1 Report

Authors have either explined / answered the questions and have improved the manuscript.

Tables are now more didatic and results more understandable.

Congratulations on your research!

Author Response

Thank you for your time reviewing this manuscript. We really appreciate that manuscript has improved from your suggestions.

We have thoroughly done proof reading.

Thank you very much.

Reviewer 2 Report

I found the revised manuscript to be much clearer than the previous version. 

Author Response

Thank you for your time reviewing this manuscript. We really appreciate that manuscript has improved from your suggestions.

We have thoroughly done proof reading.

Thank you very much.

This manuscript is a resubmission of an earlier submission. The following is a list of the peer review reports and author responses from that submission.

Round 1

Reviewer 1 Report

Interesting  article that meets the scope of the journal, clear structure, proper methodology, sound argumentation supporter with  recent bibliography,  appropriate referencing

Reviewer 2 Report

The manuscript titled "The Impact of Spousal Migration on the Mental 2 Health of Nepali Women: A Cross-Sectional Study" investigates the mental health by women whose spouses are international migrant workers.

Introduction:

Authors present a summarized but rich introduction about immigration and their concerns regarding the ill-health risk from female whose spouses are international migrant workers.

Material and Methods:

The study design was correctly described and participants randomly selected. Validated questionnaires were used, interviews and data collection were adequately performed.

Statistical procedures and softwares used were described as also information regarding a pilot study and ethical approval.

Results:

Table 1 presents socio-economic characteristics of the participants. Household income and household members are presented as a subcategory of age, please readjust (left).

Descriptives about health condition were nicely presented, at lines 142-143 a statistical significance between ages is reported but the test used to access this reported p-value was not described, was it Chi-Square?

At table 3: the informaiton regarding how the measures are presented (meand (SD)) are under BDI so that might cause confusion, please draw a line between all the three variables or change the format of the table for this information.

Discussion:

Results are presented based on the collected information and discussion about the study hypothesis are also linked with the results. A didatic and understanding discussion was presented.

Reviewer 3 Report

The Impact of Spousal Migration on the Mental Health of Nepali Women: A Cross-Sectional Study.

The study is of interest for designing health programs focused on women and their children. However, the authors need to address some issues to improve the manuscript.

Abstract

In the abstract, the authors could include a sentence justifying the relevance of the problem.

Introduction

Could the authors mention previous studies on health problems of female spouses of migrant workers? They also need to highlight the interest of the study.

Materials and methods

Please, include more details about the sampling procedure (e.g., was cluster sampling or other sampling used?) the degree of confidence, etc.

The authors need to explain why they decided to dichotomize the scores on the GHQ-12.

Results

The table 3 needs be explained. The reader needs to guess what is being compared.

I would like to know how multiple linear regression was computed.

Discussion

It would be interesting to include information about the family support that these women receive in these rural areas.

Among limitations should be added the need to analyze the number of children as a risk factor. Could the number of children be associated with the age of the women?

To what extent admitting symptoms of psychological distress can be a problem for these women?
